# The Fertility Indicator Equation Using Serum Progesterone and Urinary Pregnanediol-3-Glucuronide for Assessment of Ovulatory to Luteal Phase Transition

**DOI:** 10.3390/medicina57020134

**Published:** 2021-02-03

**Authors:** Stephen J. Usala, María Elena Alliende, A. Alexandre Trindade

**Affiliations:** 1Department of Internal Medicine, Texas Tech University Health Sciences Center, 1400 S. Coulter Street, Amarillo, TX 79106, USA; 2Programa de Cuidado y Estudio de la Fertilidad (PROCEF), Departamento de Obstetricia, Ginecología y Biología de la, Reproducción, Universidad de los Andes, Monseñor Alvaro del Portillo 12455, Santiago 7620001, Chile; melenalliende@gmail.com; 3Department of Mathematics and Statistics, Texas Tech University, 1108 Memorial Circle, Lubbock, TX 79409, USA; alex.trindade@ttu.edu

**Keywords:** progesterone, pregnanediol-3-glucuronide, PDG, natural family planning, fertility awareness methods

## Abstract

*Background and Objectives*: The Fertility Indicator Equation (FIE) has been shown to signal the fertile phase during the ovulatory menstrual cycle. It was hypothesized that this formulation, a product of two sequential normalized changes with a sign indicating direction of change, could be used to identify the transition from ovulatory to luteal phase with daily serum progesterone (P) and urinary pregnanediol-3-glucuronide (PDG) levels. *Materials and Methods*: Day-specific serum P levels from two different laboratories and day-specific urinary PDG levels from an additional two different laboratories were submitted for FIE analysis. These day-specific levels included mean or median, 5th, 10th, 90th and 95th percentile data. They were indexed to the day of ovulation, day 0, by ultrasonography, serum or urinary luteinizing hormone (LH). *Results*: All data sets showed a clear “cluster”—a periovulatory sequence of positive FIE values with a maximum. All clusters of +FIE signaled the transition from the ovulatory to luteal phase and were at least four days in length. The start day for the serum P and urinary PDG FIE clusters ranged from −3 to −1 and −3 to +2, respectively. The end day for serum P and PDG clusters went from +2 to +7 and +4 to +8, respectively. Outside these periovulatory FIE-P and FIE-PDG clusters, there were no consecutive positive FIE values. In addition, the maximum FIE-P and FIE-PDG values throughout the entire cycles were found in the clusters. *Conclusions*: FIE analysis with either daily serum P or urinary PDG levels provided a distinctive signature to recognize the periovulatory interval. The Fertility Indicator Equation served to robustly signal the transition from the ovulatory phase to the luteal phase. This may have applications in natural family planning especially with the recent emergence of home PDG tests.

## 1. Introduction

For decades, a serum progesterone(P) level has been an important part in assessing ovulatory function and the luteal phase [1] A single serum P of ≥3 ng/mL (9.54 nmol/L) has been used as an indicator of post-ovulation [2] although recent work has set a more reliable P level to confirm ovulation at 5 ng/mL (15.9 nmol/L) [3]. Serum P levels are usually < 1.5 ng/mL (4.77 nmol/L) during the follicular phase and begin to increase just before the luteinizing hormone(LH) surge [4,5]. After ovulation, there is a rapid increase in P. Minimum *p* values to confirm ovulation and luteinization have been variously cited at 1.8–5.0 ng/mL (5.72–15.90 nmol/L), [6]. Although concordance between fingerstick and venipuncture P levels has been demonstrated [7,8], no point-of-care(POC) test for blood P measurement is presently available.

There is a renewed interest in POC progesterone-based assessment of post-ovulation with the development of urine assays for the progesterone major metabolite, pregnanediol-3-glucuronide (PDG) [9,10,11,12,13,14,15,16]. Not only would a POC PDG home test aid in evaluation of fertility status, but it could also serve as an important adjunct for Natural Family Planning (NFP)/Fertility Assessment-Based Methods (FAMs) [15,16]. A role for a urinary PDG test for NFP/FAM would be to objectively signal the infertile “safe” interval and reduce the days of abstinence [14,15]. A test strip is available for urinary PDG measurement (Ovulation Double Check™, MFB Fertility, Boulder, CO, USA), but the signaling is based upon a threshold concept; there are 5 μg/mL (15.6 μmol/L) and 7 μg/mL (21.8 μmol/L) test strips [15,16]. With a PDG strip based upon this threshold algorithm, there were limitations to the sensitivity of detection during the post-ovulatory phase (82–88%) [15,16]. There was questionable reduction in the days of abstinence using the threshold method. With the 5 μg/mL (15.6 μmol/L) strip for Bouchard et al. [15], the most frequent positive test was on days 4 and 5 following the LH surge, and for Leiva et al. [16], when at least one positive PDG result was obtained, the average number of luteal days was 8.8. Importantly, 50% of the participants in one study had difficulty interpreting the PDG strips by inspection alone [16].

It was hypothesized that analysis based upon a rate-of-change function using daily PDG levels might better signal the transition from the ovulatory to luteal phase. To this end, the Fertility Indicator Equation (FIE) was employed [17]. The FIE function, as first developed, utilized a multiplicative normalized change in sequential daily serum estradiol levels with a sign rule based upon confidence in direction of change. This FIE application revealed a strong signal for the entry into the fertile range with daily serum estradiol levels. To test the applicability of the FIE for detection of the periovulatory transition, daily serum P (the standard for luteal phase assessment) and daily PDG levels from four different laboratories were submitted to FIE analysis: serum P from Stricker et al. [18] and Roos et al. [19] and urinary PDG from Johnson et al. [20] and Alliende et al. [14]. Indeed, the FIE analysis with serum P (FIE-P) and with urinary PDG (FIE-PDG) created sequences of positive FIE values—“clusters”—that occurred in the periovulatory interval denoting the transition to the luteal phase.

## 2. Methods

### 2.1. Fertility Indicator Equation Analysis

The FIE and its application to daily serum estradiol levels was detailed previously [17] and will be recapitulated here. Instead of daily serum estradiol levels, daily serum P or daily urinary PDG levels are substituted in the equation. 

Delta function values (relative rate of change in hormone level) for day “D-1” and day “D” are first calculated as
Delta_D-1_ = ((P or PDG level on D-1) − (P or PDG level on D-2))/(P or PDG level on D-2). Delta_D_ = ((P or PDG level on D) − (P or PDG level on D-1))/(P or PDG level on D-1).
where D-1 is day before D and D-2 is day before D-1. That is, three values of P or PDG are necessary to compute the FIE for Day, D. 

The magnitude of FIE on Day D is defined as:FIE (on Day D) = Delta_D-1_ X Delta_D_ X 100

A sign is then assigned to this magnitude of FIE based upon the signs of Delta_D-1_ and Delta_D_ and the following rules:**+**Delta_D-1_ X **+**Delta_D_ = **+**FIE; 
**−**Delta_D-1_ X **−**Delta_D_ = **−**FIE; 

The sign is indeterminate (ind) when

**+**Delta_D-1_ X **−**Delta_D_ = indFIE or **−**Delta_D-1_ X **+**Delta_D_ = indFIE. FIE values are therefore indeterminate if they have an indeterminate sign and are labeled “ind”.

### 2.2. Day-Specific Serum Progesterone and Urinary Pregnanediol-3-Glucuronide Levels

The day-specific P and PDG levels used for the FIE analysis are tabulated in the Appendix A. Day-specific serum P levels were used as reported by Stricker et al. [18] and Roos et al. [19], and day-specific urinary PDG levels were used as reported by Johnson et al. [20] and provided by Alliende et al. (Appendix A) [14]. These day-specific levels variously included: mean, median, 5th, 10th, 90th, 95th percentile (PCTL) levels. Neither the method to fix the day of ovulation (Day 0) nor the hormonal assays to measure P and PDG were uniform throughout these data sets. Stricker et al. used the Abbott ARCHITECT i200_SR_ system for progesterone and LH measurements, and for the FIE analysis Day 0 was indexed to the day after the LH peak. Roos et al. used the ADVIA Centaur XP Immunoassay system for P measurements and Day 0 was indexed to the day before disappearance of the dominant follicle with ultrasound. Johnson et al. used an in-house competitive immunoassay for PDG measurements (sensitivity 0.021 μg/mL (0.065 μmol/L)) and Day 0 was indexed with ultrasound.

The Alliende urinary PDG levels are from a secondary analysis of data from a multicentre World Health Organization (WHO) sponsored study (HRP#87904, approved locally, 21 September 1988). Alliende et al., used in-house non-competitive radioimmunoassays (reagents and assay protocols supplied by Matched Reagents Programme of the WHO) to measure urinary LH and PDG, and Day 0 was indexed to the day after the urinary LH rise as described previously [14].

PDG units were converted as follows: PDG μg/mL = PDG mg/L; PDG mg/L × 3.12 = PDG μmol/L.

### 2.3. Computations and Graphics

Computations and graphing for Figure 1 were performed with Microsoft Excel and GraphPad Prism version 9 for Windows, GraphPad software, San Diego, CA, USA (www.graphpad.com). The graph for Figure 2 was constructed using R, A Language and Environment for Statistical Computing (R Foundation for Statistical Computing), Vienna, Austria (https://www.R-project.org). 

## 3. Results

### 3.1. FIE Analysis of Day-Specific Serum Progesterone Provides a Luteal Phase Signature

Serum progesterone data sets from two different laboratories, Stricker et al. [18] and Roos et al. [19], were submitted for FIE computation of mean or median and 5th, 10th, 90th, and 95th percentile (PCTL) day-specific P levels. The FIE values indexed to the putative day of ovulation, Day 0, are shown in Table 1. Many of the day-specific FIE values for these progesterone data (FIE-P values) were indeterminate due to fluctuating P levels; therefore, these FIE-P values appear as ind in the tables that follow. (The complete computations with FIE-P magnitudes for the indeterminate FIE-P values are provided in the Appendix A.) Of note, the FIE analysis created clear clusters of uninterrupted consecutive positive FIE-P values (FIE-P clusters in bold, Table 1). These intervals of +FIE-P started as soon as Day −3 to Day −1. The FIE-P clusters were at least four days in length, ending on luteal days Day +2 to Day +7. Outside these periovulatory FIE-P clusters, there were no follicular phase or later luteal phase consecutive positive FIE-P values. In addition, the maximum FIE-P values were reached in these clusters. Thus, FIE analysis with daily serum progesterone levels provided a distinctive signature of increased FIE-P values during the periovulatory interval.

**Table 1 medicina-57-00134-t001:** Fertility Indicator Equation (FIE) values for day-specific serum progesterone(P) levels from Stricker et al. [18] and Roos et al. [19] using mean, median, 5th,10th, 90th and 95th percentile (PCTL) levels.

Day of Cycle (Day 0, day of ovulation) ^a^	FIE ^b^ Stricker P (mean)	FIE Stricker P (5th PCTL)	FIE Stricker P (95th PCTL)	FIE Roos P (median)	FIE Roos P (10th PCTL)	FIE Roos P (90th PCTL)
−16	--	--	--	--	--	--
−15	--	--	--	--	--	--
−14	−4.62	0	−11.35	−9.88	ind	ind
−13	−3.85	ind	ind	ind	ind	ind
−12	−1.91	0	ind	18.75	11.11	ind
−11	ind	0	ind	ind	ind	0
−10	ind	0	ind	0	ind	ind
−9	−1.62	0	ind	ind	ind	ind
−8	ind	0	ind	−1.9	ind	1
−7	ind	0	−4.44	−5.71	8.33	ind
−6	ind	0	ind	ind	0	ind
−5	0.41	0	ind	4.8	0	ind
−4	ind	0	ind	ind	ind	ind
−3	ind	0	**0.65**	ind	ind	ind
−2	**11.43**	0	**8.52**	**20.51**	**16.66**	ind
−1	**117.18**	**93.75**	**42.69**	**29.14**	**50**	**20.59**
0	**142.65**	**78.22**	**180.82**	**30.56**	**67.5**	**29.68**
1	**125.65**	**89.55**	**126.74**	**59.95**	**66.72**	**57.17**
2	**101.7**	**159.4**	**60.38**	**125.11**	**16.47**	**67.45**
3	**30.66**	**0.49**	**31.5**	**9.75**	ind	ind
4	**9.1**	**0.38**	**1.18**	**4.55**	ind	ind
5	**0.25**	ind	**0.07**	ind	ind	ind
6	**0.12**	ind	**0.46**	ind	ind	ind
7	ind	ind	**1.42**	ind	ind	ind
8	−0.24	−0.96	ind	ind	ind	−0.12
9	ind	−0.75	ind	0.02	−0.36	ind
10	ind	−1.73	ind	ind	−7.16	ind
11	−3.41	−28.24	ind	−7.02	−3.89	−9.48
12	−2.68	−18.86	ind			
13	−4.18	−4.62	ind			
14						
15						
16						

^a^ Day 0, day of ovulation, is indexed to the day after the serum LH peak for Stricker data and as detected by ultrasound for Roos data. ^b^ The FIE computations using the day-specific serum progesterone levels, the FIE-P values, are described in the Methods and listed in the Appendix A. Indeterminate FIE-P values due to indeterminant sign are indicated as ind. The bold highlights the clusters, the uninterrupted intervals of positive FIE-P.

### 3.2. FIE Analysis of Day-Specific Urinary Pregnanediol-3-Glucuronide Displays a Similar Periovulatory Cluster

Testing for the major metabolite of progesterone, PDG, in urine is now available in the form of home test strips [15,16]. We were therefore interested in FIE analysis with day-specific urinary PDG data to see if this could also enable signaling of the transition from the ovulatory to luteal phase. Two different data sets of day-specific PDG levels were used to generate FIE values (FIE-PDG values): Johnson et al. [20] and Alliende et al. (Appendix A) [14]. Indeed, observable clusters—consecutive days of positive FIE-PDG—were found (FIE-PDG clusters in bold, Table 2). These clusters began on Day −3 to Day +2 and ended on Day +4 to Day +8. The FIE-PDG clusters were ≥6 days duration. Corresponding to the FIE-P clusters, the FIE-PDG maximum values occurred within the clusters, and there were no consecutive positive FIE-PDG values outside the clusters.

**Table 2 medicina-57-00134-t002:** Fertility Indicator Equation(FIE) values for day-specific urinary pregnanediol-3-glucuronide (PDG) levels from Johnson et al. [20] and from Alliende et al. [14] using mean, median, 5th, 10th, 90th and 95th percentile(PCTL) levels.

Day of Cycle (Day 0, Day of Ovulation) ^a^	FIE Johnson PDG (Median)	FIE Johnson PDG (10th PCTL)	FIE Johnson PDG (90th PCTL)	FIE Alliende PDG (Mean)	FIE Alliende PDG (Median)	FIE Alliende PDG (5th PCTL)	FIE Alliende PDG (95th PCTL)	FIE Alliende PDG (10th PCTL)	FIE Alliende PDG (90th PCTL)
−16	--	--	--	--	--	--	--	--	--
−15	ind	ind	ind	--	--	--	--	--	--
−14	ind	ind	ind	ind	ind	ind	ind	ind	ind
−13	−0.5	0	−0.85	ind	1.37	ind	ind	ind	4.1
−12	−0.86	ind	ind	−0.23	ind	ind	ind	−0.07	ind
−11	ind	ind	ind	ind	ind	0.23	0.29	ind	ind
−10	ind	0	ind	ind	ind	ind	ind	ind	ind
−9	ind	ind	ind	−0.78	−0.09	ind	−2.58	ind	−3.18
−8	0	ind	ind	ind	ind	1.47	ind	3.75	ind
−7	ind	ind	ind	ind	ind	ind	1.57	ind	0.13
−6	ind	ind	ind	−0.26	ind	ind	ind	ind	ind
−5	0	1.78	ind	−0.58	ind	ind	−3.78	ind	−1.32
−4	0	0	−1.9	ind	ind	−0.15	ind	ind	−0.42
−3	0	0	ind	**0.43**	**0.10**	ind	**1.36**	ind	ind
−2	1.57	0	**3.14**	**0.45**	**0.51**	**0.47**	**1.6**	ind	**1.95**
−1	ind	0	**2.16**	**2.42**	**0.76**	**29.13**	**3.14**	21.53	**2.91**
0	ind	**4.54**	**8.88**	**7.91**	**4.87**	**4.36**	**9.28**	ind	**10.52**
1	**12.65**	**2.84**	**4.15**	**4.89**	**5.02**	**1.77**	**7.37**	ind	**0.1**
2	**13.54**	**3.67**	**7.57**	**15.51**	**13.73**	**13.42**	**12.37**	**57.85**	**0.24**
3	**19.67**	**28.32**	**55.07**	**32.56**	**30.50**	**55.06**	**35.05**	**26.71**	**57.64**
4	**19.23**	**7.22**	**18.43**	**15.79**	**22.38**	**7.64**	**31.58**	**12.3**	**4.5**
5	**13.33**	**2.28**	**16.35**	**1.77**	**1.06**			**13.24**	**1.46**
6	**1.55**	**1.14**	**6.74**	**0.48**	**0.19**			**0.09**	ind
7	**0.19**	**3.31**	ind	**1.79**	**1.18**			**0.01**	ind
8	ind	**8.02**	−3.79	ind	ind			ind	ind
9	ind	ind	ind	ind	ind			ind	ind
10	ind	ind	ind		ind			ind	ind
11	−5.47	ind	−4.1		−5.71			−3.79	ind
12	−2.04	ind	−5.89		−4.57			−7.05	ind
13	−1.06	ind	−2.18		−2.51			−0.24	−9.42
14									
15									
16									

^a^ Day 0 indexed according to ultrasound for Johnson et al. and by LH for Alliende et al. (ref [14] and Appendix A therein). The FIE computations using the day-specific urinary pregnanediol-3-glucuronide levels are described in the Methods and in the Appendix A. Indeterminate FIE values due to indeterminant sign are indicated by ind. The bold highlights the clusters, the uninterrupted intervals of positive FIE.

### 3.3. Characteristics and Comparison of FIE-P and FIE-PDG Clusters

Both the serum P and urinary PDG levels with FIE analysis showed a consistent pattern or signature for the transition from the ovulatory phase to the luteal phase. The mean FIE-P and FIE-PDG values with 95% confidence limits for the periovulatory interval, Day −6 to Day +6, are displayed in Figure 1. The start day, end day, day of maximum FIE value, and the maximum FIE value for the FIE-P and FIE-PDG clusters are summarized in Table 3. Plots of an averaging of the FIE-P and FIE-PDG values with minimum and maximum values are shown in Figure 2.

**Figure 1 medicina-57-00134-f001:**
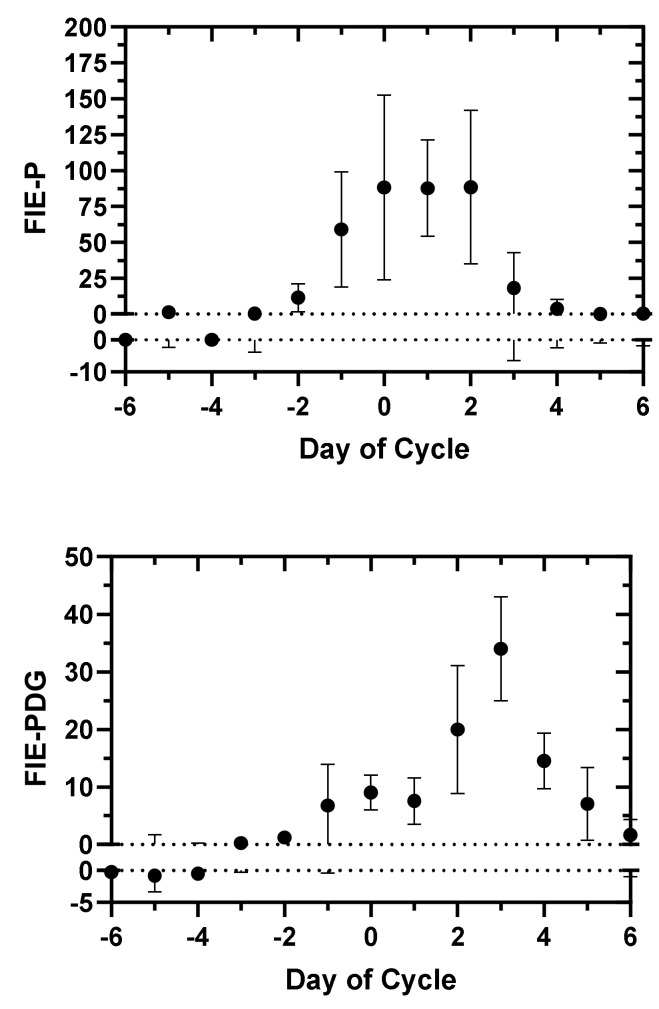
Mean Fertility Indicator Equation (FIE) values with 95% confidence limits for interval, Day −6 to Day +6, using day-specific serum progesterone levels (top) and day-specific urinary pregnanediol-3-glucuronide levels (bottom) of Stricker et al. [18], Roos et al. [19], Johnson et al. [20] and Alliende et al. (Appendix A and ref [14]). The FIE values, FIE-P and FIE-PDG, are listed in Table 1 and Table 2. Day 0, day of ovulation.

**Figure 2 medicina-57-00134-f002:**
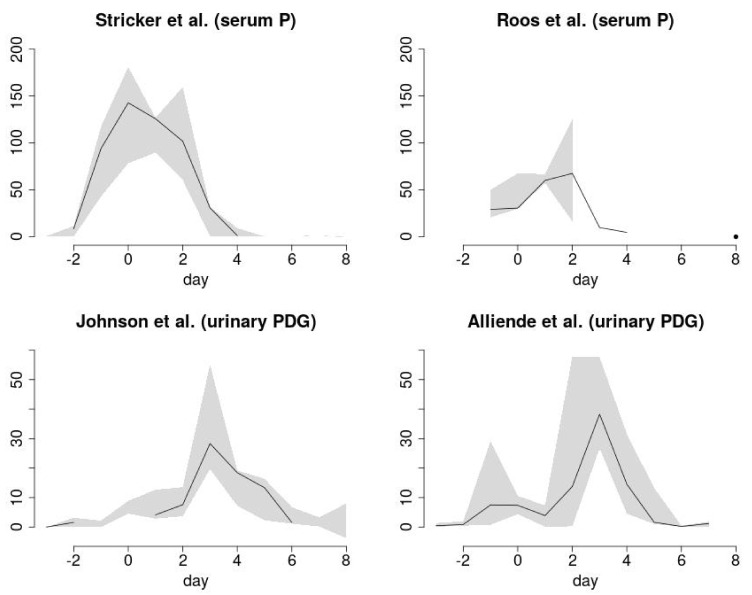
Plausible ranges of Fertility Indicator Equation (FIE) values from four different studies. The FIE values correspond to either day-specific serum progesterone (P) levels (top panels, Stricker et al. [18], Roos et al. [19], Table 1) or day-specific urinary pregnanediol-3-glucuronide (PDG) levels (bottom panels, Johnson et al. [20] and Alliende et al. (Appendix A, [14], and Table 2)), and are plotted as a function of day of cycle (Day 0, day of ovulation). Within a particular day, individual FIE values, FIE-P or FIE-PDG, were computed from all the available data in each study: typically, a low percentile, the mean or median, and a high percentile(PCTL). The Alliende data had six data sets: mean, median, 5th, 10th, 90th, 95th PCTL, all provided in Table 2. The curves display the range of FIE values within a day, with the gray area spanning the region between the extremes (minimum and maximum). Additionally, any available values between the extremes are averaged to produce the solid line. Days with no line indicate that only two FIE values are available, whereas days with only a line signify that only one FIE value is available. Single isolated FIE values (separated by blank regions) are marked with a point (e.g., day 8 in Roos et al., top right).

The FIE signatures for the urinary PDG levels had excellent concordance on the day of maximum FIE value: Day +3 with 8/9 data sets and Day +2 with 1/9 data sets. The FIE signatures with the serum P levels had maximum FIE values on Day 0 with 3/6 data sets and on Day +2 with 3/6 data sets. The FIE maximum values were higher for the serum P levels vs. the urinary PDG levels, but the FIE-PDG maximum values were significantly above 0, ≥ 19. For the FIE-P results, the highest value outside the clusters for an isolated day was ~19, well below the lowest FIE-P cluster maximum of 67. Likewise, the highest FIE-PDG value for an isolated day outside the clusters was ~4, well below the lowest FIE-PDG cluster maximum of 19. 

In summary, FIE analysis with either daily serum P or daily urinary PDG levels provided a strong signal for the transition from the ovulatory to luteal phase, often starting prior to or on the day of ovulation. 

## 4. Discussion

Serum levels of the reproductive hormones, LH, estradiol, and P have been used for decades to study fertility and treat infertility [21,22]. With the development of urinary LH, and later, urinary estrone-3-glucuronide home kits, hormonal diagnostic adjuncts became available to incorporate more traditional NFP methods [23,24,25,26,27,28,29,30]. Such advances in what is now called Fertility Assessment-Based Methods(FAMs) more recently include test strips for urinary PDG, which are based upon a threshold signal of 5–7 μg/mL (15.6–21.8 μmol/L) [15,16]. However, as discussed in the introduction, there are limitations to the readability and sensitivity of these PDG strips. 

Fertility Indicator Equation analysis provides an alternate way of determining the phase of the ovulatory cycle. FIE analysis is a function of relative change in hormonal levels in which the FIE magnitude includes a strict sign adjustment for directionality of the rate of change. Since there is a normalization factor in the FIE formulation, it is not expected to be so dependent on absolute hormone levels as is the case for the PDG strips and perhaps the E3G kits. Absolute levels can of course be valuable in assessing the phase of the cycle, but there is biological variability in these levels, especially estradiol and P [17], which puts constraints on the use of hormone-based FAM kits for the timing of sexual abstinence to avoid pregnancy. 

Thus, FIE analysis offers a different approach. As shown previously for day-specific mean and outlier estradiol levels [17], and here with day-specific mean and median and outlier serum P and urinary PDG levels, robust signatures appear which signal the fertile and luteal phases. Indeed, even with the differences in absolute levels of P and PDG due to differences in the specifications of the assays of the four laboratories, there was an unbroken sequence of positive FIE-P and FIE-PDG values—clusters—in the periovulatory interval. Some of the variation in the start date of these sequences or clusters may be due to the differences between the studies in indexing ovulation. 

The FIE-P and FIE-PDG signature raises the question: What rule would one use to determine the start day of the “safe” luteal phase, which generally occurs 1–2 days after ovulation [29,31,32]? The statistically sure answer to this question awaits FIE analysis of individual cycles and clinical application of FIE-defined algorithms. However, with the 15 FIE-P and FIE-PDG data sets here (mean, median, 5th, 10th, 90th, and 95th PCTL levels), given that the earliest start day was Day −3, and arguably assuming fertility in the 6-day interval of Day −5 to Day 0 (day of ovulation) [31,32], it is predicted that the 5th day of FIE-P or FIE-PDG rise would be the start of the infertile luteal phase. This would be the evening of Day +1 to Day +4 in 13/15 data sets and later in 2/15 data sets. Importantly, the magnitude of FIE in the cluster could also serve as a signal for the start day of the safe luteal phase. Even with some variation in the range of FIE values between the clusters, setting an FIE-PDG threshold of ≥19 one finds: in the nine FIE-PDG data sets, the start day for the safe luteal phase would be Day +3 in 8/9 data sets and Day +2 in 1/9 data sets. Thus, both a day count of positive FIE-PDG values or an FIE-PDG threshold could be used in an algorithm to determine the safe luteal period.

The application of FIE analysis for personal use, in combination with cervical/vaginal mucus, the Marquette Method, or other NFP methods/FAMs, would require a programmed home monitor with test strips for urinary PDG levels; perhaps, even relative changes in PDG band intensities could be used given the FIE formulation. PDG test strips where band intensity reflected relative urinary concentrations would be necessary. The precise number of days for the luteal cluster of positive FIE values and possible use of a FIE threshold for such a system would need to be determined in clinical trials. The efficacy for avoiding pregnancy with such a programmed “FIE meter” would need to be tested in combination with other NFP/FAMs.

## 5. Conclusions

Daily serum progesterone or daily urinary pregnanediol-3-glucuronide levels when submitted to Fertility Indicator Equation analysis reveal a sequence of positive FIE values that signal the transition from ovulatory to the luteal phase. The FIE signature created by serum P or urinary PDG levels can potentially be used for algorithms to improve NFP/FAMs.

## Figures and Tables

**Table 3 medicina-57-00134-t003:** Start and end days of the Fertility Indicator Equation(FIE) clusters, the FIE-P and FIE-PDG clusters, for Stricker and Roos day-specific serum progesterone (P) levels and Johnson and Alliende day-specific urinary pregnanediol-3-glucuronide (PDG) levels. ^a^

Data Set, Serum P or Urinary PDG	Start Day of FIE-P or FIE-PDG Cluster	End Day of FIE-P or FIE-PDG Cluster	Day of Maximum FIE-P or FIE-PDG	FIE-P or FIE-PDG Max Value
Stricker (P) mean	−2	6	0	142.65
Stricker( P) 5th PCTL	−1	4	2	159.4
Stricker (P) 95th PCTL	−3	7	0	180.82
Roos (P) median	−2	4	2	125.11
Roos (P) 10th PCTL	−2	2	0	67.5
Roos (P) 90th PCTL	−1	2	2	67.45
Johnson (PDG) median	1	7	3	19.67
Johnson (PDG) 10th PCTL	0	8	3	28.32
Johnson (PDG) 90th PCTL	−2	6	3	55.07
Alliende (PDG) mean	−3	7	3	32.56
Alliende (PDG) median	−3	7	3	30.50
Alliende (PDG) 5th PCTL	−2	4	3	55.06
Alliende (PDG) 10th PCTL	2	7	2	57.85
Alliende(PDG) 90th PCTL	−2	5	3	57.64
Alliende(PDG) 95th PCTL	-3	4	3	35.05

^a^ Stricker, Roos, Johnson, and Alliende P and PDG levels are from references (18,19,29, and 14, respectively) and tabulated in the Appendix A. Complete FIE computations for these data sets—the FIE-P and FIE-PDG values—are shown in Table 1 and Table 2, and Appendix A.

## Data Availability

Data available in Appendix A.

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
