# Peer review of "The Fertility Indicator Equation Using Serum Progesterone and Urinary Pregnanediol-3-Glucuronide for Assessment of Ovulatory to Luteal Phase Transition"

_medicina, 2021, doi:10.3390/medicina57020134_

Round 1

Reviewer 1 Report

interesting study

moderate english adjustments are required

We consider the research relevant and interesting, but some minor modifications are required. We detailed as follows :

The main question addressed by the research is the potential benefit of The Fertility Indicator Equation by determining serum progesterone and urinary pregnanediol-3-glucuronide (PGD) in order to correctly identify the transition from ovulatory to luteal phase.

The research is relevant and interesting because of the novel approach and the potential better identification of transition to "safe" luteal period compared to PGD strip test alone.

The comparison of the two methods of detection (serum P levels and urinary PGD levels) showed comparative results in the evaluation of the transition period. Also, the authors used 4 diffent laboratories for probe assessment in order to accurately identify the results.

Regarding the text, some paragraphs are more difficult to read, therefore we recommend revision of the text by a native English speaker.

In the conclusion section - the authors should detail the use of FIE-P using urinary PGD levels to improve NFP/FAMs. What do they suggest, given the self use PGD strip test is not a quantitive test (the signaling is based upon a threshold concept)? Do the suggest quantitative assessment?

Reviewer 2 Report

A novel algorithm for identifying the ovulatory to luteal transition is a welcome addition to the NFP/FAM literature.  The attempt to identify the slope of the curve of day to day changes in progesterone is an excellent possibility for identifying the end of the fertile window.

The methods could be more clearly represented - I found the formulae could be more clearly formatted.

Tables were also not ideally presented.  There was a lot of empty space.  Supplementary tables were also not clear.  While raw data can be important, presentation clarity for the audience would demand a cleaner presentation in the tables (for both the paper and supplementary tables).

The figures could have illustrated the results better.  Consider rather than scatter plot a series of sloped lines to demonstrate the FIE, with a mean slope line.  Perhaps there is a better way to demonstrate the maximum FIE as well.

In the conclusion, various suggestions are made for interpretive benefits of the FIE.  The values proposed for the safe luteal period should acknowledge that the only way this could be identified is in an effectiveness trial of this method in combination with an NFP/FAM method, i.e. chance of pregnancy based on these algorithmic numbers.  A physiologic description of what is meant by "safe luteal period" should be further clarified.  Maximum values should also be acknowledged as retrospective.  Limitations of interpretation of quantitative data by individual users of a PDG test should be acknowledged.  
